# QUANTUM-CLASSICAL KNOWLEDGE DISTILLATION VIA QUANTUM SOFT LABELS

## ABSTRACT

Quantum machine learning offers a path to leverage near-term quantum devices for tasks that remain challenging for classical models. We introduce a quantum–classical hybrid knowledge distillation framework in which variational quantum circuits, equipped with angle and Quantum Fourier Transform-inspired encodings, serve as teachers that generate expressive soft-label distributions. These signals are distilled into lightweight classical students via a hybrid loss that blends hard and soft supervision. On MNIST and CIFAR-10, students distilled from quantum teachers achieve stronger robustness to Gaussian noise and rotations than classical baselines, while retaining high clean accuracy and calibration. Crucially, this shows that even capacity-limited NISQ models can provide valuable supervisory signals, suggesting a practical route toward quantum-enhanced learning without requiring quantum inference at deployment.

## 1 INTRODUCTION

Deep neural networks have achieved remarkable success across vision, speech, and language tasks, but their rapidly increasing size poses challenges for practical deployment. To be effective outside research settings, models must balance accuracy with efficiency, robustness, and calibration Paleyes et al. (2022); Han et al. (2015); Wang (2023). Knowledge distillation (KD) provides one solution by training compact *student* networks to mimic the behavior of larger *teacher* models, thereby transferring not only predictive accuracy but also structural insights Hinton et al. (2015). Meanwhile, quantum machine learning (QML) has emerged as a promising paradigm, with evidence that even shallow quantum circuits can capture expressive representations that classical models may struggle to replicate Havlíček et al. (2019); Schuld & Killoran (2019). Yet, these two directions − classical KD and QML − have largely progressed independently, with little work seeking to unify them.

Unfortunately, most existing QML studies treat quantum models primarily as end-task classifiers, evaluated solely on accuracy over limited datasets Havlíček et al. (2019); Schuld & Killoran (2019). Yet quantum circuits are capable of generating rich probability distributions that capture uncertainty and entanglement-informed correlations Schuld & Killoran (2019); Benedetti et al. (2019). Instead of deploying quantum models directly, these distributions could be leveraged as a novel form of supervision for classical student networks, complementing the strengths of knowledge distillation with quantum-derived soft labels Hinton et al. (2015). In this way, the advantages of quantum computation may be harnessed indirectly − improving robustness, calibration, and generalization of classical networks − without necessitating quantum inference at deployment.

In this work, we introduce a quantum-classical hybrid knowledge distillation framework that turns variational quantum circuits (VQCs) Cerezo et al. (2022) into powerful soft-label teachers. By harnessing both standard angle encoding and a novel positional encoding (PE) method inspired by Quantum Fourier Transform (QFT) Coppersmith (2002), we generate expressive soft probability distributions, structure them as ordered tensors, and transfer their rich information into compact classical students such as MLPs or CNNs. Remarkably, even though the quantum teachers are weak in isolation, their outputs encode hidden structure that classical students can exploit: distilled students achieve high clean accuracy, sharply improved calibration, and dramatically enhanced robustness under Gaussian noise and rotational corruptions compared to baselines trained with conventional KD Hinton et al. (2015) or Mixup Zhang et al. (2018). Our key contributions are:

- We introduce a practical and scalable framework in which quantum teachers generate soft-label supervision for compact classical students − enabling quantum-informed learning while keeping deployment fully classical.

- We characterize how different quantum encodings induce distinct inductive biases: standard Angle Encoding preserves locality, whereas the QFT-inspired encoding (QFT-PE) imposes global spectral structure. These biases translate into complementary robustness–accuracy trade-offs across datasets.

- Through experiments on MNIST LeCun et al. (1998) and CIFAR-10 Krizhevsky (2009), we show that quantum-distilled students can outperform classical KD and Mixup under specific corruption families (e.g., rotation and Gaussian noise), even when the quantum teachers themselves are weaker than conventional models.

Taken together, these results point to a fresh paradigm for near-term quantum models: not as stand-alone classifiers, but as powerful supervisors whose rich probabilistic structure can instill enhanced robustness and calibration into compact classical networks.

**Scope of empirical evaluation:** We evaluate on MNIST and CIFAR-10 due to NISQ limits on circuit width and depth. The contribution is methodological: examining how structured quantum soft-label geometries affect robustness and calibration in classical students.

## 2 PROPOSED FRAMEWORK

### 2.1 MOTIVATION & FRAMEWORK OVERVIEW

While QML and KD have advanced independently[1], little work has explored quantum models as teachers in a distillation framework. Most quantum classifiers emphasize direct prediction rather than generating supervisory signals, leaving the potential of quantum-derived soft labels − carrying uncertainty and entanglement-informed correlations − largely untapped. This work addresses this gap by introducing a quantum–classical hybrid KD pipeline that leverages quantum soft labels to train more robust classical students.

Our goal is to bridge the representational power of quantum models with the efficiency and scalability of classical neural networks. Instead of deploying a quantum model directly as a classifier, we use it as a *teacher* that generates informative soft labels capturing richer uncertainty and class relations than one-hot targets. These quantum-derived signals are distilled into a compact classical *student*, which inherits robustness and non-classical correlations from the teacher's outputs. The distilled student not only matches or surpasses classical baselines on clean accuracy, but also shows improved calibration and robustness under corruptions compared to students trained on hard labels.

Our framework builds on two research lines. From classical KD, we adopt the transfer of softened outputs (Hinton et al., 2015), extending prior work that emphasizes richer supervisory signals (Romero et al., 2015; Zagoruyko & Komodakis, 2016; Tian et al., 2019). From quantum machine learning, we draw on advances in variational circuits and encoding strategies (Benedetti et al., 2019; Mitarai et al., 2018; Havlíček et al., 2019; Tacchino et al., 2019; Schuld & Petruccione, 2018), showing that even modest NISQ-era Preskill (2018) devices can produce expressive representations. Our framework unites these threads by positioning quantum models as soft-label generators whose outputs, distributed and merged at scale, provide classical students with quantum-informed supervision. Conceptually, it is a pipeline for *quantum-informed supervision*: quantum circuits act as label generators that encode distributional knowledge, transferred to students via distillation.

**Scalability to high-resolution datasets:** The quantum module never processes raw pixels; all images are first pooled into a fixed-length vector of dimension $n$, matching the qubit count[2]. As a result, the quantum cost is resolution-agnostic: a 224×224 ImageNet image yields the same n-dimensional input to the VQC as a 32×32 CIFAR image. Only the classical student architecture

---

[1]See details on the background and related works in Appendix A.1

[2]Although pooling reduces the dimensionality of the input, the quantum benefits in this work stem from encoding geometry rather than exponential Hilbert-space scaling, making the method stable even when $n \ll d$ and suitable for NISQ-era devices.

scales with dataset complexity. Thus, the framework is structurally compatible with ImageNet-scale pipelines, with the quantum teacher acting as a resolution-independent soft-label generator.

## 2.2 QUANTUM LABEL GENERATOR (QLG)

To extract expressive features from image data (MNIST e.g.), we use a VQC-based quantum-enhanced label generator. By average pooling, input images are downsampled and flattened into $n$ segments, where $n$ represents number of qubits. We use $n = 10$ for all experiments unless otherwise noted. To parameterize single-qubit rotations, each pooled value is mapped into the interval $[0, \pi]$. Two encoding schemes — *Angle Encoding* and a novel *QFT-PE* — are supported as interchangeable methods. Each encoding provides a different mechanism for converting pixel intensities into rotation angles before injecting them into the circuit.

**Angle Encoding:** Angle encoding provides a direct and interpretable mechanism for injecting classical information into the quantum circuit. Given a flattened input image, pixel intensities are partitioned into $n$ segments, where $n$ equals the number of qubits. Each segment is reduced by average pooling, producing $n$ scalar values in $[0, 1]$. These values are linearly mapped into the range $[0, \pi]$ and applied as rotation angles in $R_Y$ gates across the qubits:

$$\theta_i = \pi \cdot \frac{1}{|S_i|} \sum_{x \in S_i} x, \qquad i = 1, \ldots, n,$$

where $S_i$ denotes the $i$-th segment of the input. The resulting quantum state after encoding is:

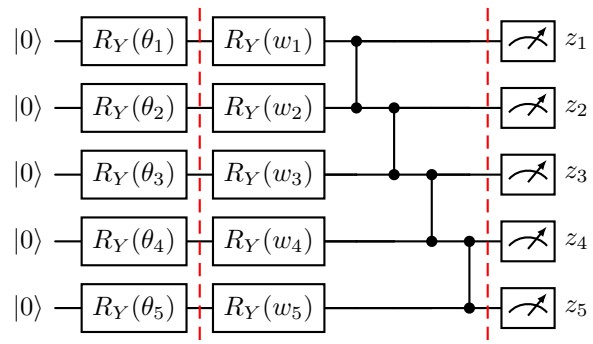

Figure 1: VQC layer: First column encodes $\{\theta_i\}$. Then one trainable layer of $R_Y(w_i)$ followed by a ring entanglement via CZ (shown for 5 qubits). Measurements yield $z_i = \langle Z_i \rangle$. For $L$ layers, repeat the $\{R_Y(w_i)\}$+CZ block between red dash-lines for $L$ times.

$$|\psi_{\text{enc}}\rangle = \bigotimes_{i=1}^{n} R_Y(\theta_i) |0\rangle,$$

This embedding ensures that each qubit receives a normalized, smooth representation of a local portion of the image.

**QFT-PE:** To explore richer positional structure in the data, we additionally implement a Quantum Fourier Transform (QFT)-inspired encoding. As in angle encoding, the image is first flattened, pooled into $n$ segments, and normalized into $[0, 1]$. However, before mapping values into $[0, \pi]$, each pooled value is modulated by its positional index $i$:

$$\theta_i = \pi \cdot \left( \left( \frac{1}{|S_i|} \sum_{x \in S_i} x \right) \cdot (i + 1) \mod 1 \right), \qquad i = 1, \ldots, n.$$

This modulation scheme, inspired by the frequency-domain structure of the QFT, introduces position-dependent phase variations into the input encoding. The resulting embedding state is:

$$|\psi_{\text{QFT-PE}}\rangle = \bigotimes_{i=1}^{n} R_Y(\theta_i) |0\rangle,$$

which allows the circuit to represent both intensity and positional information simultaneously, potentially enhancing expressivity and improving downstream classification performance. Using the selected encoding scheme, the VQC (Fig. 1) initializes qubits followed by $L$ repeated layers of parameterized $R_Y$ rotations and ring-structured controlled-$Z$ entangling gates. Specifically, each layer performs the following:

$$U(\mathbf{w}) = \prod_{i=1}^{n} R_Y(w_i) \prod_{i=1}^{n} \text{CZ}(i, (i + 1) \mod n),$$

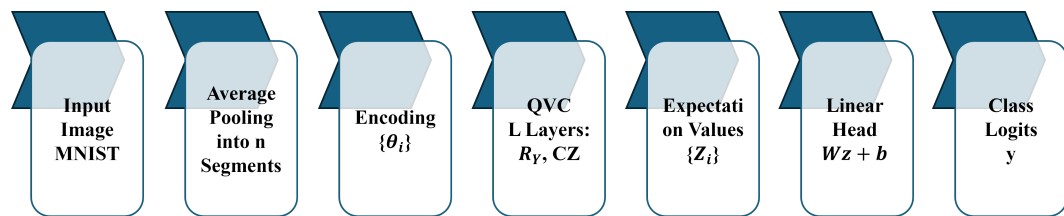

Figure 2: Pipeline of the QLG. Input images are pooled into $n$ segments, encoded, processed through a VQC, and mapped by a linear head to produce class logits.

where $\mathbf{w} = (w_1, \ldots, w_n)$ are trainable parameters shared across layers. The circuit outputs expectation values of Pauli-$Z$ operators across all qubits:

$$z_i = \langle \psi | Z_i | \psi \rangle, \qquad i = 1, \ldots, n.$$

The resulting quantum features $\mathbf{z} = (z_1, \ldots, z_n) \in \mathbb{R}^n$ are stacked into a vector of length $n$, which is passed through a linear classification head to produce logits over $C$ classes:

$$\mathbf{y} = W\mathbf{z} + \mathbf{b}, \qquad \mathbf{y} \in \mathbb{R}^C,$$

which is trained with the categorical cross-entropy loss:

$$\mathcal{L} = -\sum_{c=1}^{C} \mathbf{y}_c^* \log \mathsf{softmax}(\mathbf{y})_c.$$

This allows for end-to-end optimization with standard losses such as cross-entropy. Weight sharing across circuit layers is applied to reduce parameter count and stabilize training, though the implementation allows for unique per-layer weights if needed. The workflow is depicted in Fig. 2.

Encoding choice acts as an explicit inductive bias: QFT-PE emphasizes global spectral structure, whereas Angle Encoding preserves locality. This makes the framework adaptable rather than fragile − different tasks require different priors, just as classical architectures differ in convolutional vs. attention-based bias. The MNIST–CIFAR-10 contrast simply reflects the alignment between encoding geometry and dataset structure.

**NISQ compatibility:** The VQC uses 5 layers with $n \leq 10$ qubits, ring entanglement, and single-qubit rotations, which fits within the depth and connectivity limits of current NISQ hardware. The quantum teacher is run offline to generate soft labels, and deployment remains entirely classical, making the pipeline hardware-feasible even with present-day devices. Importantly, the design deliberately sets $n \ll d$ (e.g., 8–10 qubits for MNIST), not to exploit exponential Hilbert-space dimensionality, but to ensure stable optimization and tractable circuit depth. The robustness effects studied in this work stem from the geometry of the encoding − global spectral structure in QFT-PE and locality preservation in Angle Encoding − combined with entanglement-induced mixing across qubits, rather than from representing all input dimensions in quantum space.

## 2.3 Distributed Soft-Label Generation (DDP)

We generate temperature-scaled *soft labels* for MNIST using a pretrained VQC-based QLG (Sec. 2.2). The dataset is partitioned across $W$ processes via a deterministic `DistributedSampler`; each rank $r$ consumes a disjoint index set $\mathcal{I}_r$ (no shuffle). For a batch $\{\mathbf{x}^{(j)}\}_{j=1}^B$ the model yields logits $\mathbf{y}^{(j)} \in \mathbb{R}^C$, converted to probabilities with temperature $T > 0$:

$$p_c^{(j)} = \frac{\exp(\mathbf{y}_c^{(j)}/T)}{\sum_{k=1}^C \exp(\mathbf{y}_k^{(j)}/T)}, \quad c = 1, \ldots, C. \tag{1}$$

Each rank writes a shard containing $\{$`split`, `indices`, `probs`, $T$, $r$, $W\}$; an all-ranks barrier ensures completion. Here, `split` $\in \{$`train`, `test`$\}$ denotes the dataset partition being processed; it is distinct from DDP Li et al. (2020) rank sharding. Indices stored in each shard are global within this split. We use analytic expectation values (`shots=None`) and eval-only inference; per-example cost is $O(nL)$ with near-linear wall-time scaling in $W$.

## 2.4 MERGING DISTRIBUTED SOFT-LABEL SHARDS

We merge per-rank soft-label shards produced by DDP into a single ordered tensor for each dataset partition ($\text{split} \in \{\text{train}, \text{test}\}$). We first verify consistent world size across shards and equal local shard sizes (MNIST is divisible by the chosen world size). Let $W$ be the world size, $r$ the rank, and $i$ the row index within a shard; the global row index is

$$j \;=\; r \;+\; i\,W,$$

and we place each local probability row at position $j$ to reconstruct the original dataset order. The merged tensor is subsequently saved. As a sanity check, we compute top-1 accuracy by comparing $\arg\max$ over the merged probabilities with the ground-truth MNIST labels for the same split.

## 2.5 CLASSICAL STUDENT TRAINING VIA KNOWLEDGE DISTILLATION

We train a compact student classifier on MNIST using precomputed quantum soft labels (Sec. 2.3). The script supports two architectures: a 2-layer MLP over flattened $28{\times}28$ pixels and a small CNN (two conv blocks, $2\times$ max-pooling, and a 256-unit head). Data transforms include optional robustness augmentations (random rotation, additive Gaussian noise, random erasing) and MNIST-stat normalization; augmentations are applied only to the training split.

**Inputs and dataset wrapper:**  Soft labels are loaded from ordered tensors saved per split as a dictionary with key $\text{probs} \in \mathbb{R}^{N \times C}$; lengths are checked to match the torchvision MNIST order. A dataset wrapper returns triplets (image, $y_{\text{hard}}$, $\mathbf{p}_{\text{teacher}}$), where $\mathbf{p}_{\text{teacher}}$ may be None.

**Loss − hard CE + KD:**  We minimize a hybrid loss combining categorical cross-entropy on hard labels (with optional label smoothing $\varepsilon$) and a KL term that matches the student's temperature-scaled distribution to the teacher's probabilities:

$$\tilde{\mathbf{y}} = (1 - \varepsilon)\,\mathsf{onehot}(y) + \frac{\varepsilon}{C}\,\mathbf{1}, \qquad \mathcal{L}_{\text{CE}} = -\sum_{c=1}^{C} \tilde{\mathbf{y}}_c \log \mathsf{softmax}(\mathbf{z}_s)_c,$$

$$\mathcal{L}_{\text{KD}} = T^2\,\mathsf{KL}\Big(\mathbf{p}_{\text{teacher}} \,\Big\|\, \mathsf{softmax}(\mathbf{z}_s/T)\Big), \qquad \mathcal{L} = \alpha\,\mathcal{L}_{\text{CE}} + (1 - \alpha)\,\mathcal{L}_{\text{KD}}.$$

Here $\mathbf{z}_s$ are student logits, $T > 0$ is the KD temperature (applied to the student; teacher probs are assumed normalized), and $\alpha \in [0, 1]$ trades off hard vs. soft supervision. During evaluation the KD term is disabled ($\mathbf{p}_{\text{teacher}}$=None).

**Training setup and checkpointing.**  We optimize with Adam (learning rate as $2{\times}10^{-3}$), mini-batch size (default 256), and train for a user-set number of epochs. The script reports train/test loss and accuracy each epoch and saves the best checkpoint (by test accuracy) to student_ckpt/ with the selected model type in the filename. Random seeds are fixed for reproducibility.

Per batch, the forward/backward cost is $\mathcal{O}(BP)$ where $P$ is the number of student parameters (one of MLP or CNN); KD adds only a small $\mathcal{O}(BC)$ overhead for temperature scaling and KL.

## 3 RESULTS AND DISCUSSIONS

Unless noted, results are mean $\pm$ std over three seeds; single-run tables are marked.

## 3.1 SANITY, CALIBRATION, AND CORRUPTION CHECKS

We perform four checks to validate the pipeline (see details in Appendix A.2: **(1)** Shard integrity and ordering by verifying DDP-generated soft-label shards for consistent world size, row sums $\approx 1$, sane value range and no NaN/Inf. **(2)** Calibration: accuracy, NLL, ECE (15 bins), Brier; bin-wise confidence/accuracy for reliability plots. **(3)** Corruptions: rotation, noise, translation, contrast with severities $s \in \{0, 0.25, 0.5, 0.75, 1.0\}$; report per-severity metrics and AUC-Acc. **(4)** Controls on MNIST: head-only (frozen quantum) and label-shuffled to rule out leakage.

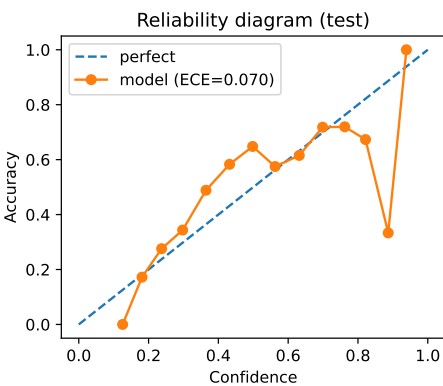

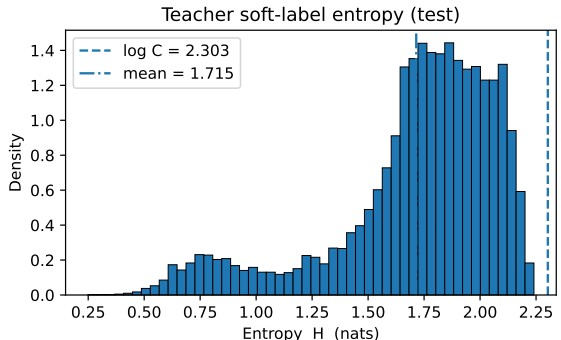

Figure 3: Reliability diagram (test split)    Figure 4: Entropy histogram of teacher soft labels on `test`. Dashed line: $\log 10 \approx 2.303$

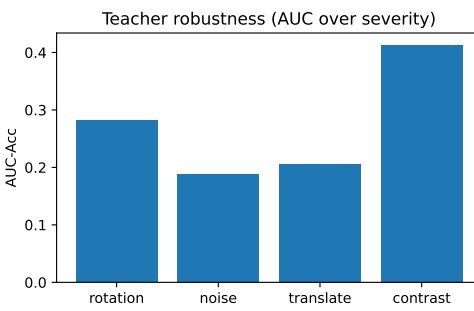

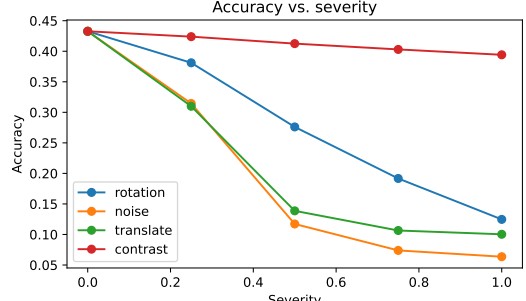

Figure 5: Teacher robustness. Left: AUC-Acc per corruption family (higher is better). Right: Accuracy vs. severity curves for each family.

## 3.2 CALIBRATION AND ROBUSTNESS OF THE TEACHER

For the four sanity checks enumerated above, the following is what we have observed (interested readers are directed to Appendix A.2 for details): **(1)** Both `train` and `test` shard demonstrate consistent `world_size`, row sums $\approx 1$, sane value range, and no NaN/Inf. **(2)** On `test`, the teacher attains 43.27% accuracy, 1.625 NLL, 0.070 ECE, and 0.0718 Brier, Table 1. The reliability bins reveal under-confidence in the mid-range. Figure 3 confirms this on `test` (15 bins), yielding ECE $= 0.070$, which indicates mild under-confidence in the mid-confidence regime. **(3)** Accuracy declines most under *noise* and *translation*, is moderate under *rotation*, and is least affected by *contrast*. AUC-Acc: rotation 0.282, noise 0.188, translation 0.205, contrast 0.413, Table 2 and Fig. 5 reaffirm these findings. **(4)** Head-only reaches 34.63% acc (NLL 1.937, ECE 0.107, Brier 0.0806), well above chance, while label-shuffled: 14.03% acc with NLL 2.312 (close to $\ln 10 \approx 2.303$), ECE 0.024, and Brier 0.0902, provide evidence of no leakage and correct metric wiring.

Table 1: Teacher calibration on `test`

|         | Acc (%) | NLL ↓ | ECE ↓ | Brier ↓ |
|---------|---------|-------|-------|---------|
| Teacher | 43.27   | 1.625 | 0.070 | 0.0718  |

Table 2: Robustness (AUC-Acc over severities)

| Corruption | Rot.  | Noise | Tran. | Cont. |
|------------|-------|-------|-------|-------|
| AUC-Acc    | 0.282 | 0.188 | 0.205 | 0.413 |

## 3.3 SOFT-LABEL AUDIT: ORDERING, ACCURACY, ENTROPY, AND QUALITATIVE SAMPLES

To verify the integrity and informativeness of the teacher's soft labels, we first reconstruct the globally ordered probability matrix $\mathbf{P} \in \mathbb{R}^{N \times C}$ from $W$ DDP shards. Each shard $r \in \{0, \ldots, W-1\}$ contributes a local block $\mathbf{P}^{(r)} \in \mathbb{R}^{N_{\mathrm{loc}} \times C}$; we place row $i$ of shard $r$ at global index $j = r + iW$, which yields a contiguous ordering without overlap. On the corresponding dataset split (train or

Table 3: Teacher soft-label quality on `test`. $H$ is in nats; $H_{\max} = \log 10 \approx 2.303$.

| Split | $N$ | Acc@Argmax (%) | $\mathbb{E}[H]\downarrow$ | $\mathrm{Std}[H]$ | $\min H$ | $\max H$ |
|---|---|---|---|---|---|---|
| `test` | 10,000 | 43.27 | 1.7149 | 0.3722 | 0.2486 | 2.2406 |

Table 4: Teacher vs. student comparison on MNIST

| Model | Acc (%) $\uparrow$ | NLL $\downarrow$ | ECE $\downarrow$ | Brier $\downarrow$ |
|---|---|---|---|---|
| Teacher (QFT) | 43.40 | 1.8130 | 0.025 | 0.6534 |
| Student (QFT KD, $T = 2$) | $96.85 \pm 0.14$ | $0.42 \pm 0.00$ | $0.28 \pm 0.00$ | $0.17 \pm 0.00$ |

test), we compute (i) *accuracy* of $\arg\max_c \mathbf{P}_{n,c}$ against ground-truth labels, and (ii) the *entropy* of each soft label reported in nats: $H_n = -\sum_{c=1}^{C} \mathbf{P}_{n,c} \log \mathbf{P}_{n,c}$.

Table 3 summarizes accuracy and entropy statistics; Fig. 4 shows the entropy distribution with a vertical line at $\log C$. The entropy distribution is well below $\log 10$, with substantial spread, indicating informative (non-uniform) soft labels (details in Appendix). Argmax accuracy corroborates teacher quality on `test`. We use these soft labels unchanged for KD unless otherwise stated (e.g., temperature sweeps). We visualize a small set of images with ground-truth label, predicted class, and top-$k$ probabilities to qualitatively assess whether high- and low-entropy cases match human intuition, Fig. 7 (in Appendix A.3). Low-entropy samples show confident, human-aligned predictions (e.g. $GT = 7 \rightarrow P(7) = 0.94$), while high-entropy cases are ambiguous, validating the teacher's uncertainty encoding. This sanity test gives a quick end-to-end check for (a) correct shard reassembly, (b) plausibility of teacher accuracy on the target split, and (c)

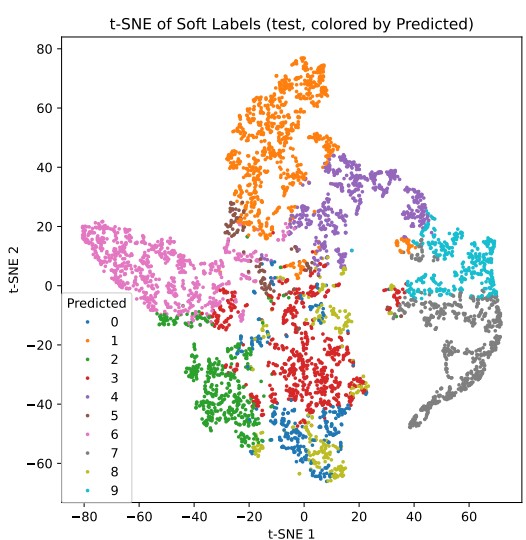

Figure 6: Trustworthiness via t-SNE: Predicted

whether the soft labels meaningfully reflect uncertainty (e.g., $H$ near 0 for easy cases and near $\log C$ for ambiguous cases). Finally, we visualize the geometry of the teacher's soft labels by applying t-SNE van der Maaten & Hinton (2008) to the probability matrix $\mathbf{P} \in \mathbb{R}^{N \times C}$ (rows are samples; columns are class probabilities). We report the embedding trustworthiness as a sanity metric and provide colorings, either by the teacher's predicted class (argmax) or by ground-truth, in Figure 6 (details in Appendix A.3).

### 3.4 QUANTUM TEACHER TO CLASSICAL STUDENT KNOWLEDGE DISTILLATION

Table 4 compares the QFT teacher and its student: despite the limited accuracy of NISQ-era models, QFT-KD yields a 96.9% student with solid calibration, illustrating effective transfer.

### 3.5 CALIBRATION-ACCURACY TRADEOFF

See Appendix A.4 for temperature sweep results showing improved calibration with T = 2.

### 3.6 ROBUSTNESS TO GAUSSIAN NOISE AND ROTATIONS - CLASSICAL VS. HYBRID

We evaluate the final *student* models on MNIST under a grid of input corruptions that combine Gaussian noise and in-plane rotations. For experimental settings, see Appendix A.5. Table 5 shows

Table 5: Robustness of quantum–classical methods on MNIST

| | QFT KD (student) | | | | | Angle KD (student) | | | | |
|---|---|---|---|---|---|---|---|---|---|---|
| $\sigma$ | 0° | 15° | 30° | 45° | 60° | 0° | 15° | 30° | 45° | 60° |
| 0.0 | 96.50 | 95.64 | 92.98 | 76.10 | 50.08 | 94.78 | 93.94 | 91.11 | 72.38 | 43.78 |
| 0.1 | 93.49 | 90.98 | 83.00 | 62.84 | 37.80 | 90.57 | 88.64 | 80.24 | 58.12 | 34.36 |
| 0.2 | 93.47 | 90.75 | 83.20 | 62.71 | 37.75 | 90.41 | 88.38 | 79.97 | 58.04 | 34.00 |
| 0.3 | 93.42 | 90.83 | 83.11 | 62.71 | 37.87 | 90.10 | 88.24 | 79.39 | 57.65 | 33.58 |
| 0.4 | 93.43 | 90.80 | 83.27 | 62.70 | 37.94 | 89.80 | 87.78 | 79.06 | 57.24 | 33.46 |
| 0.5 | 93.43 | 90.79 | 83.07 | 63.18 | 37.91 | 89.65 | 87.47 | 78.61 | 56.94 | 33.32 |
| 0.6 | 93.42 | 90.85 | 83.48 | 63.16 | 38.07 | 89.36 | 87.21 | 78.21 | 56.87 | 33.19 |
| 0.7 | 93.23 | 90.84 | 83.48 | 63.98 | 38.54 | 89.18 | 86.71 | 77.66 | 56.26 | 32.91 |
| 0.8 | 93.34 | 90.98 | 83.66 | 63.94 | 38.77 | 88.99 | 86.33 | 77.29 | 55.79 | 32.93 |
| 0.9 | 93.57 | 91.09 | 84.18 | 65.12 | 39.51 | 88.72 | 86.07 | 77.01 | 55.97 | 33.01 |

that the student distilled from the QFT-encoded teacher (QFT KD) is consistently more robust than the Angle KD student at every rotation: for example at $0°/60°$ (no noise) it attains $96.50/50.08$ vs. $94.78/43.78$, a margin of $+1.7/+6.3$ points. This advantage persists when combining noise and rotation (e.g., at $\sigma=0.9$, $45°$: $65.12$ vs. $55.97$; at $60°$: $39.51$ vs. $33.01$). Notably, the QFT KD student shows only modest accuracy loss under Gaussian noise at $0°$ ($\approx 96.5\% \to 93.6\%$ as $\sigma_{\text{noise}}$ rises to $0.9$), while strong degradation appears only with large rotations.

**Classical baselines:** Results for Classical KD and Mixup (LeNet backbone) under identical corruption settings are reported in Appendices A.6, A.7, A.8 and A.9. They achieve high clean accuracy but degrade sharply under rotations and combined noise–rotation stress, providing a context for interpreting hybrid quantum–classical robustness.

To separate the effect of *quantum* structure from generic low-confidence supervision, we compared QFT-PE against two classical controls: a Fourier-based teacher operating on low-frequency FFT coefficients, and a deliberately weak CNN teacher matched to the quantum teacher's accuracy. Distillation from the Fourier teacher improved robustness but remained significantly below QFT-KD, indicating that frequency bias alone does not explain the effect. Distillation from the matched-accuracy weak CNN produced no robustness benefit, showing that robustness does not arise from teacher weakness but from the structure of the quantum soft-label geometry.

For CIFAR-10, the same compact student architecture that worked well on MNIST was retained without scaling up. On this more complex dataset, all students: quantum-distilled and classical show constrained absolute accuracy. Nevertheless, the quantum-distilled students still outperform purely classical baselines in robustness, while within the quantum variants, Angle KD demonstrates a relative advantage over QFT KD, Table 6. We discuss this contrast in the following paragraph. Meanwhile, for additional clarity, we report difference tables ($\Delta$ accuracy in percentage points) in Appendix A.7 and A.8 (Tables 12 and 13). These highlight the margins by which quantum-distilled students surpass Classical KD and Mixup under noise and rotations on CIFAR-10, reinforcing the robustness advantage observed in the main results. Additional comparisons with augmentation-only classical students are reported in Appendix A.8, showing that improvements from QFT-KD are not attributable to extra augmentation.

**QFT KD vs. Angle KD** On MNIST, QFT-KD yields higher clean accuracy and stronger robustness, reflecting its global spectral bias. On CIFAR-10, Angle-KD performs better under corruption due to its locality $-$ preserving inductive bias. See Table 7.

## 3.7 LIMITATIONS OF QFT KD ROBUSTNESS

While QFT-KD delivers clear gains on Gaussian noise and rotations, the robustness is not uniform across corruption types (see Appendix A.10) $-$ we do not claim that these cover all practically relevant perturbations. The improvements arise from the encoding-induced inductive bias rather than a universal invariance: QFT-PE encourages global, low-frequency structure that aligns naturally with

Table 6: Robustness of quantum–classical methods on CIFAR-10

| $\sigma$ | QFT KD (student) | | | | | Angle KD (student) | | | | |
|---|---|---|---|---|---|---|---|---|---|---|
| | 0° | 15° | 30° | 45° | 60° | 0° | 15° | 30° | 45° | 60° |
| 0.00 | 65.0 | 57.1 | 43.0 | 26.3 | 22.7 | 63.4 | 59.1 | 50.8 | 36.6 | 30.0 |
| 0.10 | 66.8 | 61.7 | 47.5 | 29.3 | 24.9 | 63.1 | 60.0 | 51.8 | 37.0 | 30.1 |
| 0.20 | 49.4 | 46.3 | 36.7 | 25.3 | 20.7 | 52.3 | 50.5 | 43.6 | 32.3 | 27.2 |
| 0.30 | 34.4 | 30.9 | 26.1 | 21.4 | 19.5 | 40.6 | 39.5 | 33.1 | 25.7 | 23.1 |
| 0.40 | 25.7 | 23.2 | 20.8 | 18.6 | 16.9 | 32.6 | 30.9 | 25.7 | 22.1 | 20.0 |
| 0.50 | 20.5 | 18.2 | 17.0 | 15.8 | 14.7 | 27.5 | 26.2 | 22.4 | 19.3 | 18.3 |
| 0.60 | 16.9 | 15.9 | 14.7 | 13.8 | 13.3 | 24.3 | 22.8 | 19.9 | 18.1 | 17.0 |
| 0.70 | 14.2 | 14.5 | 13.1 | 12.4 | 11.9 | 21.6 | 21.1 | 19.2 | 17.4 | 16.6 |
| 0.80 | 13.7 | 13.0 | 12.6 | 12.1 | 11.8 | 20.6 | 19.5 | 18.1 | 16.8 | 16.0 |
| 0.90 | 13.1 | 12.2 | 11.8 | 11.6 | 11.2 | 19.2 | 18.3 | 17.3 | 16.4 | 15.8 |

Table 7: Clean accuracy, mean robustness (mean over $\sigma \in \{0, \cdots, 0.9\}$ and rotations $\{0°, 15°, 30°, 45°, 60°\}$), and retention (mean/clean).

| Dataset / Teacher | | Clean (%) | Mean grid (%) | Retention |
|---|---|---|---|---|
| MNIST | QFT | 96.50 | 74.70 | 0.774 |
| MNIST | Angle | 94.78 | 70.21 | 0.741 |
| CIFAR-10 | QFT | 65.00 | 24.20 | 0.372 |
| CIFAR-10 | Angle | 63.40 | 29.42 | 0.464 |

rotation and structured noise, whereas translation and contrast shifts require local spatial consistency better matched by Angle Encoding. Thus the framework offers controllable, encoding-specific robustness rather than blanket distributional invariance. Future work includes combining multiple encodings or hybridizing with classical augmentations to achieve broader robustness. Extending the framework to adversarial perturbations and broader distribution shifts also remains important future work but lies beyond the scope of the present study.

## 4 CONCLUSION

This work introduced a quantum–classical hybrid knowledge distillation framework where weak variational quantum teachers provide soft labels that enhance the robustness of compact classical students. Across MNIST and CIFAR-10, we found that QFT-based encodings deliver stronger geometric robustness than classical baselines, while Angle encoding balances clean accuracy with broader stability. The results highlight that quantum models, even when limited in standalone performance, can serve as valuable supervisors for classical networks. Our approach allows training with quantum teachers while preserving fully classical inference, making it attractive for deployment in resource-constrained settings. These findings suggest a practical role for quantum-informed supervision in building lightweight models that are both accurate and resilient to corruptions, pointing to a promising direction for hybrid machine learning research.

## USE OF THE LARGE LANGUAGE MODELS (LLMS)

While the authors wrote the technical content, designed and executed all experiments, and interpreted the results, LLM assistance was employed to improve clarity and brevity, particularly when fitting the paper to the 9-page limit. Specifically, polishing was applied to abstract, introduction (contribution bullets), related work gap statements, stylizing equations, few selected sentences in results, and conclusion. No scientific claims, analyses, or results were generated by an LLM.

## ETHICS STATEMENT

This work aligns with the ICLR Code of Ethics by contributing methods that support responsible, transparent, and reproducible machine learning research for knowledge-distillation-based image classification. All data sets used for this research are publicly available and duly cited, whereas no human subjects, private data, or personally identifiable information were used.

## REPRODUCIBILITY STATEMENT

All experiments were run with fixed random seeds, and dataset preprocessing steps, training configurations, and evaluation metrics are fully described in the paper and Appendix. To support reproducibility, we will release the complete source code (including Jupyter notebooks, Python scripts, trained checkpoints, and evaluation results), along with the curated dataset and a README for detailed guidance, either during the rebuttal phase upon request or with the camera-ready submission.

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

# A APPENDIX

## A.1 BACKGROUND AND BASELINES

### A.1.1 CLASSICAL KNOWLEDGE DISTILLATION

Knowledge distillation (KD) emerged as a solution to the growing computational and memory demands of deep neural networks. By transferring knowledge from a large teacher model to a smaller student, KD enables model compression for deployment on resource-constrained devices while often improving generalization through richer supervision than one-hot labels provide. Beyond efficiency, KD also offers a way to expose the inductive biases and learned structure of high-capacity models, making it a versatile tool for both practical deployment and representation learning.

Hinton et al. Hinton et al. (2015) introduced the seminal KD framework, where students are trained to match the softened output distributions of teachers using a temperature-scaled softmax. This objective, which combines hard-label cross-entropy with a KL-divergence loss, remains the foundation of most distillation approaches. Several representative extensions build on this idea: FitNets Romero et al. (2015) transfer intermediate feature "hints" to guide deeper, thinner students; Attention Transfer Zagoruyko & Komodakis (2016) aligns spatial attention maps between teacher and student; and Contrastive Representation Distillation (CRD) Tian et al. (2019) employs contrastive learning to preserve structural dependencies in representations. While not strictly a KD method, Mixup Zhang et al. (2018) creates interpolated samples and labels to provide soft targets, and is frequently used as a strong comparative baseline in distillation studies.

Taken together, these works illustrate a progression in KD research: from transferring softened outputs, to leveraging intermediate representations, to aligning structural and relational information—broadening the scope and effectiveness of knowledge transfer.

In recent years, KD has been pushed well beyond its original goal of compressing large models. A growing line of work shows it can also help students become better calibrated and more robust. Guo et al. Guo et al. (2017) pointed out that modern networks tend to be miscalibrated, sparking interest in temperature scaling and distillation methods that account for confidence. Born-Again Networks Furlanello et al. (2018) went a step further, showing that even when teacher and student have the same size, repeating the distillation process can actually make the student stronger than the teacher. Other efforts have focused on robustness, finding that distilled students can inherit resilience to adversarial perturbations and distribution shifts from their teachers Goldblum et al. (2020); Yuan et al. (2020). These studies suggest that KD is not just a way to shrink models, but also a practical tool for building models that are more reliable in the wild.

Beyond early soft-label distillation, several lines of work have explored robustness- and calibration-aware KD, including feature-level transfer Wang et al. (2025), attention-based transfer, and contrastive representation distillation Bao et al. (2025). These methods highlight how different teacher geometries can shape student robustness and uncertainty. Our approach is complementary: instead of designing new classical teachers, we investigate small quantum teachers with structured encodings as a source of soft-label geometry under NISQ constraints.

### A.1.2 QUANTUM MACHINE LEARNING FOR SUPERVISED TASKS

**Quantum-native Models** The adaptation of supervised learning to quantum computing has motivated a growing body of work on how quantum circuits can process classical data, learn predictive mappings, and potentially achieve quantum advantage Schuld & Killoran (2022); Macaluso et al. (2023). These efforts provide both theoretical foundations and practical strategies for hybrid pipelines that combine quantum and classical models Broughton et al. (2020).

Schuld et al. Schuld & Petruccione (2018) systematically framed supervised learning in a quantum setting, introducing quantum encodings, feature maps, and learning objectives that cast quantum circuits as extensions of classical learners. Havlíček et al. Havlíček et al. (2019) advanced this perspective by proposing quantum feature maps that embed classical data into high-dimensional Hilbert spaces, enabling kernel-based classification and demonstrating quantum kernels on real devices. Building on these foundations, Benedetti et al. Benedetti et al. (2019) surveyed parameterized

quantum circuits as expressive learning models, highlighting encoding strategies, representational capacity, and practical challenges such as barren plateaus and hardware noise.

Together, these contributions establish the foundations of quantum supervised learning, progressing from theoretical frameworks, to encoding and kernel methods, to expressive parameterized circuits that balance quantum capabilities with near-term hardware constraints.

**Quantum-Classical Hybrid Models**     Quantum neural models generalize classical neural networks to the quantum domain, leveraging quantum parallelism and entanglement for expressive learning while staying practical on near-term devices. They typically employ variational circuits and hybrid training schemes that adapt machine learning methods to quantum hardware. While most prior QML studies deploy quantum models directly as classifiers, recent perspectives emphasize alternative supervisory roles. Jerbi et al. Jerbi et al. (2023) discuss quantum learning beyond kernels, and early work has also explored distillation-like quantum–classical interactions Abbas et al. (2021). Our work builds on these insights, treating quantum models as soft-label teachers rather than end-task predictors.

Mitarai et al. Mitarai et al. (2018) introduced Quantum Circuit Learning, a framework combining low-depth variational circuits with classical optimization for supervised tasks. They showed that such circuits can approximate nonlinear functions and, with sufficient qubits, act as universal function approximators, while retaining a training process analogous to backpropagation. Building on this, Tacchino et al. Tacchino et al. (2019) implemented a quantum perceptron on IBM quantum devices, demonstrating exponential storage advantages and providing one of the earliest experimental validations of quantum neural classifiers. More recently, Jerbi et al. Jerbi et al. (2023) examined the scalability of variational quantum circuits in the NISQ era, emphasizing architectural choices and resource-efficient strategies to maintain learning performance under hardware constraints.

Together, these works chart the evolution of quantum neural models from theoretical formulations to hardware demonstrations, and toward scalable architectures designed to overcome the limitations of current quantum devices.

## A.2  Sanity-check and Observation Details

1. We check shard integrity and ordering by verifying DDP-generated soft-label shards for (i) consistent `world_size`, (ii) per-row probability normalization, value range, and NaN/Inf checks, and (iii) total row count.

2. We compute calibration metrics including accuracy, negative log-likelihood (NLL), expected calibration error (ECE, 15 bins), and Brier score; we also store bin-wise average confidence and accuracy for reliability diagrams.

3. We observe corruptions and robustness by evaluating four corruption families with severities $s \in \{0, 0.25, 0.5, 0.75, 1.0\}$ respectively for *rotation*, *noise*, *translation*, and *contrast*. For each family we report per-severity metrics and the area under the accuracy–severity curve (AUC-Acc).

4. We run two controls on `MNIST` ($C{=}10$, chance $= 10\%$). (1) *Head-only (frozen quantum):* we freeze all quantum circuit parameters and train only the linear head for 3 epochs (Adam, lr $10^{-3}$, batch 128) on the training set, then evaluate on `test`. (2) *Label-shuffled:* we train the full model under the same settings, but permute labels within each batch at every step. This breaks input–label alignment while keeping class frequencies, so any above-chance test accuracy would indicate leakage or a bug.

For the four sanity checks enumerated above, the following is what we have observed (interested readers are directed to Appendix A.2 for details):

1. Both `train` and `test` shard demonstrate consistent `world_size`, row sums $\approx 1$, sane value range, and no NaN/Inf. For 10-qubit encodings over 2,000 samples we observe mean angle $\approx 1.16$ rad, std $\approx 0.99$ rad, with $\sim 23\%$ of angles near 0 and $< 1\%$ near $\pi$ (indicative of a mild small-angle pile-up).

2. On `test`, the teacher attains 43.27% accuracy, 1.625 NLL, 0.070 ECE, and 0.0718 Brier, Table 1. The reliability bins reveal under-confidence in the mid-range, where average ac-

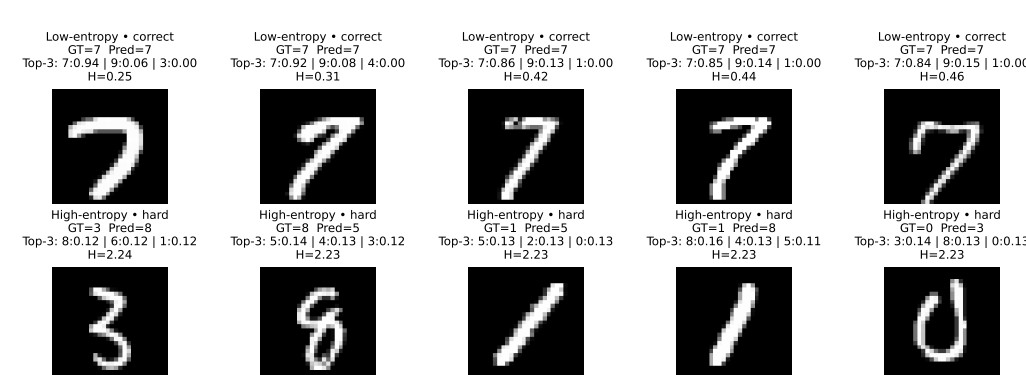

Figure 7: Qualitative audit: each panel shows *GT*, predicted class, and top-3 probabilities. We display both low-entropy (confident) and high-entropy (ambiguous) cases.

curacy exceeds average confidence (e.g., $\hat{p} \approx 0.36$ vs. acc $\approx 0.49$; $\hat{p} \approx 0.43$ vs. acc $\approx 0.58$). Figure 3 shows the reliability diagram on `test` (15 bins). The curve sits close to the diagonal and yields ECE = 0.070, indicating mild under-confidence in the mid-confidence regime and only slight over-confidence near the highest bin.

3. Accuracy declines most under *noise* and *translation*, is moderate under *rotation*, and is least affected by *contrast*. AUC-Acc (higher is better): rotation 0.282, noise 0.188, translation 0.205, contrast 0.413, Table 2. The bar chart in Fig. 5 (left) shows that *contrast* is least harmful, followed by *rotation*, while *translation* and especially *Gaussian noise* degrade accuracy more sharply. The line plot (right) confirms the monotone accuracy drop with severity and highlights the relative ordering across families. From the underlying results, the AUC-Acc values are: contrast 0.413, rotation 0.282, translation 0.205, and noise 0.188.

4. Results reveal that head-only reaches 34.63% acc (NLL 1.937, ECE 0.107, Brier 0.0806), well above chance, confirming the quantum-embedded features carry usable signal even with random circuit weights. Label-shuffled yields 14.03% acc with NLL 2.312 (close to $\ln 10 \approx 2.303$), ECE 0.024, and Brier 0.0902, i.e., near-uniform predictions and near-chance accuracy—evidence of no leakage and correct metric wiring.

## A.3 Soft-Label Audit Details

We visualize a small set of images with ground-truth label, predicted class, and top-$k$ probabilities to qualitatively assess whether high- and low-entropy cases match human intuition, Fig. 7. This sanity test gives a quick end-to-end check for (a) correct shard reassembly, (b) plausibility of teacher accuracy on the target split, and (c) whether the soft labels meaningfully reflect uncertainty (e.g., $H$ near 0 for easy cases and near $\log C$ for ambiguous cases).

We visualize the geometry of the teacher's soft labels by applying t-SNE to the probability matrix $\mathbf{P} \in \mathbb{R}^{N \times C}$ (rows are samples; columns are class probabilities). We subsample $m$ points uniformly at random (for speed), fit a 2D t-SNE embedding (perplexity 30, `init=pca`, fixed random seed), and color points either by the teacher's predicted class (argmax) or by ground-truth. This plot is purely qualitative: well-formed, class-coherent clusters and small "islands" aligned with plausible confusions (e.g., $2 \leftrightarrow 3$) indicate informative soft labels, whereas diffuse clouds would point to near-uniform outputs. We report the embedding trustworthiness as a sanity metric and provide both colorings in Figure 8.

## A.4 Calibration-Accuracy Tradeoff Table

On MNIST, $T=1$ maximizes accuracy; $T=2$ improves calibration with minor accuracy cost; $T=4$ over-smooths. See Table 8.

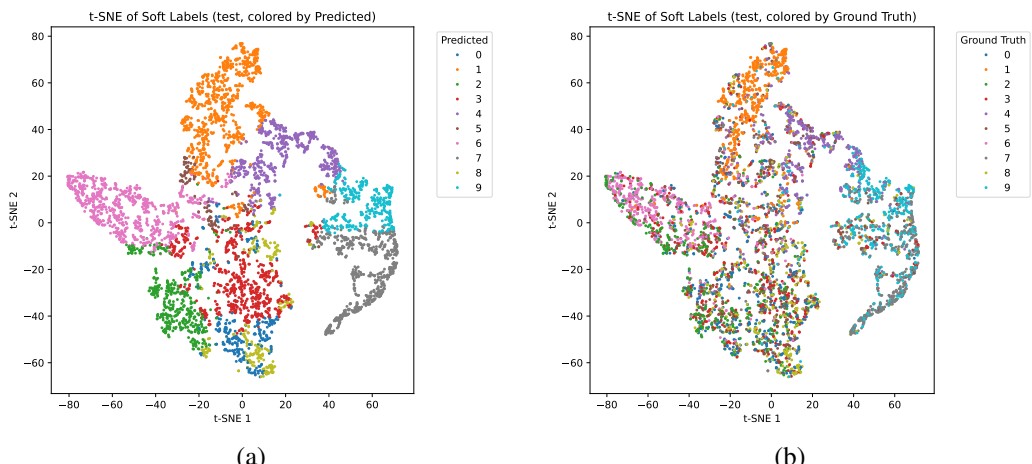

(a)          (b)

Figure 8: Embedding trustworthiness via t-SNE: (a) Ground-truth, (b) Predicted

Table 8: QFT KD (student CNN): mean $\pm$ std over 3 seeds for each temperature $T$

| $T$ | Acc (%) $\uparrow$ | NLL $\downarrow$ | ECE $\downarrow$ | Brier $\downarrow$ | $n$ |
|---|---|---|---|---|---|
| 1 | $98.89 \pm 0.03$ | $0.48 \pm 0.03$ | $0.32 \pm 0.05$ | $0.19 \pm 0.01$ | 3 |
| 2 | $96.85 \pm 0.14$ | $0.42 \pm 0.00$ | $0.28 \pm 0.00$ | $0.17 \pm 0.00$ | 3 |
| 4 | $89.46 \pm 0.82$ | $0.42 \pm 0.00$ | $0.28 \pm 0.00$ | $0.16 \pm 0.00$ | 3 |

### A.5 SETTINGS FOR ROBUSTNESS TO NOISE & ROTATION EXPERIMENTS

As in training, the test images are normalized with the dataset statistics $(\mu, \sigma) = (0.1307, 0.3081)$. Given a normalized input $x \in \mathbb{R}^{1 \times 28 \times 28}$, we form

$$x' = \mathrm{clamp}\Big(\mathrm{Rot}_\theta(x) + \sigma_{\mathrm{noise}} \cdot \sigma \cdot \varepsilon, -1, 1\Big), \qquad \varepsilon \sim \mathcal{N}(0, \mathbf{I}),$$

with $\theta \in \{0°, 15°, 30°, 45°, 60°\}$ (filled background $= 0$) and $\sigma_{\mathrm{noise}} \in \{0.0, 0.1, \ldots, 0.9\}$. We report test accuracy (%) for two quantum–classical students distilled from VQC teachers $-$ QFT KD and Angle KD $-$ and compare against two classical baselines (Classical KD and Mixup, both with LeNet backbones). All models are evaluated in `eval` mode with no test-time augmentation. $\alpha - sensitivity$ : We varied the CE/KD weight $\alpha \in \{0.3, 0.5, 0.7\}$ for Classical KD. Across these settings, test accuracy fluctuated by less than 0.2% and calibration metrics varied negligibly. We therefore fixed $\alpha = 0.5$ throughout all other experiments.

### A.6 CLASSICAL KD & MIXUP BASELINES ON MNIST

The classical baselines (Table 9), Classical KD and Mixup, achieve the highest clean accuracies (98.74% and 98.90% at 0°, $\sigma{=}0$), but are substantially *less* rotation-robust: at $45°/60°$ and $\sigma{=}0$ they reach $56.17/33.05$ and $57.28/34.41$.

### A.7 CLASSICAL KD WITH CIFAR-10

For classical knowledge distillation (KD), we use a moderately deeper convolutional teacher composed of two $3{\times}3$ convolutional layers (with 32 and 64 filters, padding 1), followed by a max-pooling layer, a fully connected layer with 128 units, and a 10-way output layer. The student is a LeNet model consisting of two convolutional layers (6 and 16 filters, kernel size 5), followed by three fully connected layers with dimensions 120, 84, and 10. All models are trained for 50 epochs using the Adam optimizer (learning rate $10^{-3}$, batch size 64). KD is implemented using a soft label loss that interpolates between cross-entropy and KL divergence with temperature $T{=}2.0$ and blending factor $\alpha{=}0.5$.

Table 9: Robustness of classical methods on MNIST

| | Classical KD (LeNet) | | | | | Mixup (LeNet) | | | | |
|---|---|---|---|---|---|---|---|---|---|---|
| $\sigma$ | 0° | 15° | 30° | 45° | 60° | 0° | 15° | 30° | 45° | 60° |
| 0.0 | 98.74 | 96.91 | 84.13 | 56.17 | 33.05 | 98.90 | 96.76 | 84.31 | 57.28 | 34.41 |
| 0.1 | 98.55 | 96.45 | 83.36 | 54.85 | 31.89 | 97.26 | 93.14 | 78.46 | 50.27 | 30.40 |
| 0.2 | 98.04 | 95.43 | 81.14 | 52.92 | 29.73 | 90.78 | 85.05 | 69.49 | 42.54 | 27.12 |
| 0.3 | 96.60 | 93.32 | 77.61 | 49.55 | 28.41 | 80.26 | 72.96 | 56.05 | 35.51 | 23.23 |
| 0.4 | 93.26 | 88.82 | 71.37 | 45.70 | 26.89 | 67.65 | 57.61 | 44.04 | 28.08 | 19.00 |
| 0.5 | 86.27 | 81.55 | 63.96 | 41.16 | 25.04 | 52.71 | 44.02 | 32.74 | 22.30 | 16.15 |
| 0.6 | 77.64 | 72.73 | 56.25 | 36.76 | 23.46 | 40.30 | 33.89 | 25.85 | 18.40 | 14.50 |
| 0.7 | 67.21 | 63.46 | 49.06 | 32.85 | 21.64 | 31.89 | 26.79 | 21.44 | 16.31 | 12.46 |
| 0.8 | 58.51 | 53.92 | 42.76 | 29.00 | 19.62 | 26.28 | 22.79 | 19.33 | 14.54 | 12.15 |
| 0.9 | 50.59 | 46.70 | 37.16 | 27.21 | 18.38 | 21.87 | 20.73 | 16.92 | 13.56 | 11.16 |

To evaluate robustness, we test the student model on grayscale CIFAR-10 (resized to $28 \times 28$) under increasing levels of Gaussian noise and affine rotation. As shown in Table 10, the KD model achieves a clean test accuracy of 63.36%. Compared to Mixup (Table 11), the KD-trained student exhibits stronger resilience to moderate perturbations. For instance, the model maintains over 50% accuracy at $15°$ rotation and retains over 23% accuracy under Gaussian noise with standard deviation 0.2. This suggests that classical KD may transfer not only predictive capacity but also partial robustness from the teacher model, even when the student is low-capacity.

Table 10: Test accuracy (%) of a LeNet student trained via classical knowledge distillation on grayscale CIFAR-10 ($28 \times 28$), evaluated under increasing Gaussian noise and rotation

| Noise | 0° | 15° | 30° | 45° | 60° |
|---|---|---|---|---|---|
| 0.0 | 63.36 | 51.65 | 35.04 | 21.96 | 19.87 |
| 0.1 | 26.98 | 22.91 | 17.88 | 14.09 | 12.78 |
| 0.2 | 23.38 | 20.01 | 16.13 | 13.24 | 11.92 |
| 0.3 | 19.55 | 18.33 | 15.06 | 13.11 | 12.36 |
| 0.4 | 16.99 | 15.76 | 13.87 | 12.31 | 11.80 |
| 0.5 | 15.38 | 14.06 | 12.65 | 11.35 | 11.71 |
| 0.6 | 13.38 | 12.81 | 11.91 | 11.19 | 11.46 |
| 0.7 | 12.19 | 12.16 | 11.53 | 11.29 | 11.14 |
| 0.8 | 11.70 | 11.57 | 11.56 | 11.08 | 11.44 |
| 0.9 | 11.34 | 11.21 | 10.64 | 10.50 | 10.94 |

## A.8 MIXUP WITH CIFAR-10

For Mixup experiments, we use a classic LeNet architecture consisting of two convolutional layers (6 and 16 filters, kernel size 5), followed by three fully connected layers with sizes 120, 84, and 10. Mixup is applied with interpolation strength $\alpha = 0.4$. The model is trained on grayscale CIFAR-10 (resized to $28 \times 28$) for 10 epochs using the Adam optimizer.

To assess robustness, we evaluate the trained model under Gaussian noise and affine rotations. Table 11 reports the results. While the model achieves a clean accuracy of 63.77%, performance degrades sharply under perturbations. For example, a $15°$ rotation drops accuracy to 45.58%, and Gaussian noise with standard deviation 0.2 reduces accuracy below 16%. These findings suggest that Mixup improves generalization under clean conditions but does not confer meaningful robustness to distributional shifts when used with shallow architectures like LeNet.

**Classical augmentation baseline:** To contextualize the robustness gains, we trained a student using only classical augmentations − Gaussian noise, random rotations, and label smoothing − under the same training budget as QFT-KD. This augmentation-only baseline improves over a vanilla student but remains weaker than QFT-KD both in mean robustness over the noise–rotation grid

Table 11: Test accuracy (%) of Mixup-trained LeNet on grayscale CIFAR-10 ($28\times28$) under varying levels of Gaussian noise and rotation

| $\sigma$ | 0° | 15° | 30° | 45° | 60° |
|---|---|---|---|---|---|
| 0.0 | 63.77 | 45.58 | 30.96 | 19.71 | 19.20 |
| 0.1 | 24.28 | 17.76 | 14.71 | 11.82 | 10.92 |
| 0.2 | 15.95 | 13.23 | 11.71 | 10.88 | 10.65 |
| 0.3 | 12.41 | 11.40 | 10.79 | 10.39 | 10.08 |
| 0.4 | 11.32 | 10.54 | 10.34 | 10.14 | 10.01 |
| 0.5 | 10.41 | 10.24 | 10.19 | 9.91 | 10.00 |
| 0.6 | 10.03 | 9.97 | 9.98 | 10.01 | 9.97 |
| 0.7 | 10.04 | 10.07 | 9.99 | 9.98 | 9.92 |
| 0.8 | 10.00 | 9.99 | 10.02 | 9.98 | 10.01 |
| 0.9 | 10.00 | 10.00 | 10.00 | 10.00 | 10.01 |

Table 12: $\Delta$ accuracy (pp) vs **Classical KD** on CIFAR-10 under Gaussian noise ($\sigma$) and rotation. Left block: **QFT KD (student)**−Classical KD; Right block: **Angle KD (student)**−Classical KD.

| | QFT KD − Classical KD | | | | | Angle KD − Classical KD | | | | |
|---|---|---|---|---|---|---|---|---|---|---|
| $\sigma$ | 0° | 15° | 30° | 45° | 60° | 0° | 15° | 30° | 45° | 60° |
| 0.00 | 1.64 | 5.45 | 7.96 | 4.34 | 2.83 | 0.04 | 7.45 | 15.76 | 14.64 | 10.13 |
| 0.10 | 39.82 | 38.79 | 29.62 | 15.21 | 12.12 | 36.12 | 37.09 | 33.92 | 22.91 | 17.32 |
| 0.20 | 26.02 | 26.29 | 20.57 | 12.06 | 8.78 | 28.92 | 30.49 | 27.47 | 19.06 | 15.28 |
| 0.30 | 14.85 | 12.57 | 11.04 | 8.29 | 7.14 | 21.05 | 21.17 | 18.04 | 12.59 | 10.74 |
| 0.40 | 8.71 | 7.44 | 6.93 | 6.29 | 5.10 | 15.61 | 15.14 | 11.83 | 9.79 | 8.20 |
| 0.50 | 5.12 | 4.14 | 4.35 | 4.45 | 2.99 | 12.12 | 12.14 | 9.75 | 7.95 | 6.59 |
| 0.60 | 3.52 | 3.09 | 2.79 | 2.61 | 1.84 | 10.92 | 10.00 | 8.00 | 6.91 | 5.54 |
| 0.70 | 1.99 | 2.34 | 1.57 | 1.11 | 0.76 | 9.41 | 8.94 | 7.67 | 6.11 | 5.46 |
| 0.80 | 2.00 | 1.43 | 1.04 | 1.02 | 0.36 | 8.90 | 7.93 | 6.54 | 5.72 | 4.56 |
| 0.90 | 1.76 | 0.99 | 1.16 | 1.10 | 0.26 | 7.86 | 7.09 | 6.66 | 5.90 | 4.86 |

and in calibration (higher ECE). These results indicate that the observed gains do not arise from additional augmentation but from the structure of the quantum teacher's soft-label geometry.

## A.9 MULTI-SEED STABILITY OF CLASSICAL KD, MIXUP AND QFT KD

We report a multi-seed Classical KD baseline (student: LeNet, $T{=}2.0$) on MNIST, Table 14. Over 3 seeds, the Classical KD student achieves $99.05{\pm}0.10\%$ accuracy with strong calibration (NLL

Table 13: $\Delta$ accuracy (pp) vs **Mixup** on CIFAR-10 under Gaussian noise ($\sigma$) and rotation. Left block: **QFT KD (student)**−Mixup; Right block: **Angle KD (student)**−Mixup.

| | QFT KD − Mixup | | | | | Angle KD − Mixup | | | | |
|---|---|---|---|---|---|---|---|---|---|---|
| $\sigma$ | 0° | 15° | 30° | 45° | 60° | 0° | 15° | 30° | 45° | 60° |
| 0.00 | 1.23 | 11.52 | 12.04 | 6.59 | 3.50 | -0.37 | 13.52 | 19.84 | 16.89 | 10.80 |
| 0.10 | 42.52 | 43.94 | 32.79 | 17.48 | 13.98 | 38.82 | 42.24 | 37.09 | 25.18 | 19.18 |
| 0.20 | 33.45 | 33.07 | 24.99 | 14.42 | 10.05 | 36.35 | 37.27 | 31.89 | 21.42 | 16.55 |
| 0.30 | 21.99 | 19.50 | 15.31 | 11.01 | 9.42 | 28.19 | 28.10 | 22.31 | 15.31 | 13.02 |
| 0.40 | 14.38 | 12.66 | 10.46 | 8.46 | 6.89 | 21.28 | 20.36 | 15.36 | 11.96 | 9.99 |
| 0.50 | 10.09 | 7.96 | 6.81 | 5.89 | 4.70 | 17.09 | 16.00 | 12.21 | 9.39 | 8.29 |
| 0.60 | 6.87 | 5.93 | 4.72 | 3.79 | 3.33 | 14.27 | 12.83 | 9.92 | 8.09 | 7.03 |
| 0.70 | 4.16 | 4.43 | 3.11 | 2.42 | 1.98 | 11.56 | 11.03 | 9.21 | 7.42 | 6.68 |
| 0.80 | 3.70 | 3.01 | 2.58 | 2.12 | 1.79 | 10.60 | 9.51 | 8.08 | 6.82 | 5.99 |
| 0.90 | 3.10 | 2.20 | 1.80 | 1.60 | 1.19 | 9.20 | 8.30 | 7.30 | 6.40 | 5.79 |

Table 14: Classical KD, Mixup vs. QFT KD on MNIST ($T$=2.0). Mean $\pm$ std over 3 seeds

| Method | Acc (%) ↑ | NLL ↓ | ECE ↓ | Brier ↓ |
|---|---|---|---|---|
| Classical KD (LeNet) | 99.05 $\pm$ 0.10 | 0.0319 $\pm$ 0.0040 | 0.0039 $\pm$ 0.0009 | 0.0153 $\pm$ 0.0020 |
| Mixup (LeNet) | 99.03 $\pm$ 0.08 | 0.0607 $\pm$ 0.0046 | 0.0329 $\pm$ 0.0051 | 0.0188 $\pm$ 0.0009 |
| Proposed (QFT) | 96.85 $\pm$ 0.14 | 0.42 $\pm$ 0.00 | 0.28 $\pm$ 0.00 | 0.17 $\pm$ 0.00 |

0.0319$\pm$0.040, ECE 0.0039$\pm$0.0009, Brier 0.0153$\pm$0.0020), while the Mixup student achieves competitive figures as well: 99.03$\pm$0.08% accuracy with calibration (NLL 0.0607$\pm$0.046, ECE 0.0329$\pm$0.0051, Brier 0.0188$\pm$0.0009). By contrast, the QFT-distilled student (row 3) is slightly weaker in clean accuracy but substantially more robust under geometric corruptions, underscoring the robustness–accuracy trade-off.

## A.10  LIMITATIONS VIA ADDITIONAL CORRUPTIONS

Beyond Gaussian noise and rotation, we also evaluate student robustness to translation (6-pixel shift) and contrast reduction (factor 0.2). Classical KD students, despite strong clean accuracy (99.05% $\pm$ 0.10), perform poorly under translation (4.1% $\pm$ 1.1) and show unstable behavior under contrast (87.1% $\pm$ 14.5). Mixup (LeNet) achieves similarly high clean accuracy (99.03% $\pm$ 0.08) but generalizes weakly to translation (12.5% $\pm$ 0.9) and shows inconsistent contrast robustness (72.8% $\pm$ 14.8). The QFT-distilled student (CNN, $T$=2.0) maintains strong clean accuracy (96.9%) but performs poorly under translation (4.5%) and contrast reduction (19.6%). As summarized in Table 15, these results indicate that the robustness benefits of quantum distillation are highly *corruption-specific*: QFT KD excels on rotations and Gaussian noise but does not extend this advantage to spatial shifts or contrast changes. These findings emphasize that high clean accuracy does not guarantee robustness across distribution shifts, motivating future work on student models that inherit broader robustness properties from their teachers.

Table 15: Additional robustness of distilled students under translation and contrast corruptions (mean $\pm$ std over 3 seeds).

| Method | Translation (px=6) | Contrast (factor=0.2) |
|---|---|---|
| Classical KD (LeNet) | 4.14 $\pm$ 1.10 | 87.10 $\pm$ 14.54 |
| Mixup (LeNet) | 12.52 $\pm$ 0.94 | 72.83 $\pm$ 14.79 |
| QFT KD (CNN) | 4.49 $\pm$ 0.00 | 19.58 $\pm$ 0.00 |

## A.11  QUBIT-COUNT ABLATION

To assess sensitivity to circuit width, we trained VQC teachers with 8, 9, 10 qubits under identical settings (MNIST 10k, 20 epochs, 5-layer circuit). Test accuracies are given in Table 16.

The results exhibit the characteristic non-monotonicity of NISQ variational circuits: 8-qubit models lack expressivity, while 10-qubit models suffer from harder optimization (barren-plateau effects). The 9-qubit model provides the best trade-off and was used for all KD experiments.

Table 16: Qubit Count Ablation

| Qubits | 8 | 9 | 10 |
|---|---|---|---|
| Acc (%) | 41.54 | 43.56 | 40.02 |

