# OpenReview forum: "Quantum-Classical Knowledge Distillation via Quantum Soft Labels"
_ICLR.cc/2026/Conference — Submitted to ICLR 2026_

### Official Review · Reviewer_kSbH · 2025-10-24

**Soundness:** 1
**Presentation:** 1
**Contribution:** 1
**Rating:** 0
**Confidence:** 4

**Summary:**

The authors propose to distil the knowledge from quantum teachers into classical models. They claim that although these quantum teachers perform poorly in isolation, their improved robustness to noise can be transferred to the student models without incurring any accuracy drop.

**Strengths:**

Attempting to combine two very disparate fields is interesting and can bring new insights.

**Weaknesses:**

I am not familiar with the quantum ML literature and so cannot make any comments on how this papers fits in there. However, I can comment on the practical utility and strengths/weaknesses with regards to the KD literature.

The datasets are very small to be of any practical benefit. It is unclear if any of these results would scale to larger models or datasets. In fact, the results on CIFAR10 are very low. This may be ok if there are some very significant theoretical contributions, but I am doubtful. I have followed the maths and it is mostly definitions describing the methodology proposed.

Is robustness to gaussian noise even a practically useful problem? how does the proposed quantum distillation compare to training for longer and with gaussian noise being injected into the data? This would be the most basic baseline compared to that is missing. In fact a more useful setting would be adversarial robustness, but it seems this property is not inherent in quantum models.

It is well known that distillation inherits various properties from the teacher models. This makes the fact that the classical models inherit some level of robustness to be very not surprising. There is almost no related work discussion from the knowledge distillation literature, except the very early works from nearly a decade ago.

**Questions:**

see weaknesses

---

### Official Review · Reviewer_TWxy · 2025-10-29

**Soundness:** 3
**Presentation:** 3
**Contribution:** 3
**Rating:** 6
**Confidence:** 4

**Summary:**

This study presents a quantum–classical hybrid knowledge distillation framework where weak variational quantum models act as teachers, providing soft labels to improve the robustness of compact classical student models. The paper presents two encoding strategies: angle-angle-based and QFT-based encodings. Experiments on MNIST and CIFAR-10 show that QFT-based encodings offer superior geometric robustness, while angle encoding achieves a balance between accuracy and stability. In addition, the paper shows that using the quantum classical hybrid improves the robustness of the Student model against Gaussian noise and rotations.

**Strengths:**

The paper studies an interesting avenue that uses QML models as a teacher for a classical model for knowledge distillation. Therefore, instead of directly using quantum models for predictions, in this work, the advantages of quantum computation may be harnessed indirectly − improving robustness, calibration, and generalization of classical networks − without necessitating quantum inference at deployment.

The paper provides a nice overview of the problem, motivations, and the proposed techniques. The numerical results clearly show the robustness of the quantum classical paradigm against Gaussian noise and rotation

**Weaknesses:**

A major limitation of the proposed method lies in its approach to encoding classical data into quantum states. Specifically, the technique flattens images into nnn segments, which effectively bypasses the exponential dimensionality advantage typically offered by quantum systems. This raises concerns about the scalability of the method. For instance, if the image size increases significantly beyond those used in the paper, and the number of segments nnn remains small relative to the image size, it is unclear whether the model would retain its robustness compared to classical counterparts.

This issue is particularly important because the computational advantages of quantum systems generally emerge only when operating with a large number of qubits. Without leveraging the full expressive power of quantum Hilbert spaces, the proposed method risks offering limited benefits over classical models, especially in high-dimensional or large-scale scenarios.

**Questions:**

Can you comment on the robustness when $n$ is significantly smaller than the image size?
Have you considered other encoding techniques, such as amplitude encoding, that potentially use the exponential dimensionality of qubits?

---

### Official Review · Reviewer_WKuB · 2025-11-01

**Soundness:** 2
**Presentation:** 3
**Contribution:** 2
**Rating:** 4
**Confidence:** 4

**Summary:**

This paper proposes a quantum-classical knowledge distillation framework. A Variational Quantum Circuit (VQC), acting as a teacher, generates soft labels. These quantum soft labels, generated using Angle or a novel QFT-inspired encoding, are distilled into a classical student network. The central claim is that this process enhances the student's robustness to Gaussian noise and rotations compared to classical baselines, even when the VQC teacher itself has low accuracy.

**Strengths:**

+ The core idea of using a (weak) quantum model as a supervisor or soft-label generator for a classical student is a novel and practical approach to leveraging NISQ devices. It cleverly bypasses the need for quantum inference at deployment time.
+ The paper demonstrates a significant and impressive improvement in robustness to rotations and combined noise on MNIST. The QFT-distilled student (39.51%) dramatically outperforms strong classical baselines like Classical KD (18.38%) and Mixup (11.16%) under the hardest corruption settings.
+ The introduction of the QFT-inspired positional encoding (QFT-PE) is an interesting contribution. The paper provides a plausible analysis of its inductive bias (favoring low-frequency features) and why this bias helps on MNIST but not CIFAR-10.
+ The study is well-structured. It includes important sanity checks (e.g., teacher performance, shuffled-label controls) and compares against strong, relevant classical baselines (Classical KD and Mixup).

**Weaknesses:**

- The central weakness is that the claimed "enhanced robustness" is not general. As admitted in Section 3.7 and Appendix A.8, the QFT-distilled student performs worse than classical baselines on translation and contrast corruptions . This suggests the method does not confer general robustness, but rather swaps one inductive bias (e.g., translation invariance) for another (rotation invariance). This significantly overstates the central claim.
- The paper does not successfully isolate the "quantum" contribution. The VQC teacher is very simple (10 qubits, basic entanglement) and extremely weak (43% accuracy on MNIST). The robustness gains appear to stem entirely from the classical QFT-PE encoding, which imparts a strong low-frequency bias. A purely classical teacher with a similar Fourier-based bottleneck or bias might achieve the same result. This alternative is not tested.
- The quantum teacher (43% acc) is compared to a strong classical teacher (LeNet, >98% acc). The paper frames this as a positive (distilling from a weak teacher), but it's an uncontrolled comparison. It's possible that any very weak, low-confidence teacher (classical or quantum) produces better soft labels for robustness than a high-confidence one. This possibility is not explored.
- The experiments are limited to 10 qubits. More importantly, the relative performance of the two quantum encodings (QFT-PE vs. Angle) completely flips between MNIST and CIFAR-10, suggesting the results are highly sensitive to the specific dataset and the chosen encoding's bias, rather than a general property of "quantum soft labels."

**Questions:**

1.	Given that the robustness gains are highly specific (strong on rotation, weak on translation/contrast), how can this be presented as a general "robustness" improvement rather than a trade-off or a specialized inductive bias?
2.	Have you attempted to replicate the robustness gains using a purely classical teacher that incorporates a similar Fourier-based bias (e.g., a strong classical model trained on data with high-frequency components removed)? How can you be sure the advantage is "quantum" and not just an artifact of the (classical) QFT-PE encoding?
3.	The quantum teacher is exceptionally weak (43% acc) while the classical teacher is strong (>98%). Have you tried distilling from a classically weak teacher (e.g., a tiny MLP trained to ~43% acc)? Is it possible that the robustness gain comes from distilling from any low-confidence teacher, rather than a specifically quantum one?
4.	The QFT-PE encoding is superior on MNIST, but Angle encoding is superior on CIFAR-10. This suggests the choice of encoding is dataset-specific and critical. Does this fragility not undermine the idea of a general "quantum-classical distillation" framework?

---

### Official Review · Reviewer_ojn2 · 2025-11-04

**Soundness:** 2
**Presentation:** 3
**Contribution:** 2
**Rating:** 2
**Confidence:** 3

**Summary:**

This paper proposes a hybrid knowledge distillation (KD) framework that leverages variational quantum circuits as teachers to supervise lightweight student models through quantum-generated soft labels.

**Strengths:**

1. The paper is well written with clear logical flow and mathematical definitions.
2. The idea of using quantum models as teachers for classical students in a KD framework is interesting.

**Weaknesses:**

1. The evaluation is not comprehensive. All experiments are conducted on small-scale datasets (MNIST, CIFAR-10). It will be better if large datasets like ImageNet can be included.
2. The authors claim that the method is attractive for deployment in resource-constrained setting. It will be better if more evidence can be provided.
3. It will be better if ablations across qubit counts can be included.

**Questions:**

1. How do you expect the framework to apply to more complex datasets, e.g., ImageNet?
2. Can this approach realistically operate on current NISQ hardware?
3. The robustness study focuses on Gaussian noise and rotation corruptions. Have the authors tested against other real-world perturbations, such as adversarial noise or domain shifts?

---

### Meta-Review · Area_Chair_1DtX · 2026-01-07

**Summary:**

This paper looks at an interesting idea, using small quantum models as teachers to guide classical models through soft-label distillation. The motivation is clear and the experiments are carefully done, especially in the NISQ setting. However, the results are mostly on small datasets, and the robustness gains depend heavily on the specific encoding choices. Overall, the work shows promise, but it would benefit from broader validation and stronger evidence of impact, so it is not accepted at this time.

**Reviewer Concerns:**

The rebuttal addressed many clarity and feasibility concerns, including scalability, NISQ compatibility, encoding choices, and several missing baselines raised by Reviewers ojn2, WKuB, and TWxy. In particular, the authors clarified robustness claims and added controlled comparisons to disentangle weak-teacher and Fourier effects. However, concerns about limited datasets, lack of large-scale validation, and the perceived incremental nature of the contribution, especially from Reviewer kSbH, remain only partially resolved.

**Reviewer Scores:**

Reviewer ojn2 would likely increase slightly from a reject to a borderline score after the added clarifications on scalability and qubit ablations. Reviewer WKuB would likely remain near the original borderline score, as the rebuttal resolves several technical concerns but leaves questions about general robustness. Reviewer TWxy would likely stay around a marginal accept, with concerns largely addressed. Reviewer kSbH is unlikely to change substantially, as their core skepticism about novelty and practical relevance remains.

---

### Decision · Program_Chairs · 2026-01-26

Reject